# High Performance Broadcast Receiver Based on Obsolete Technology

**DOI:** 10.3390/s22186784

**Published:** 2022-09-08

**Authors:** Laurenţiu Teodorescu, Gabriel Dima

**Affiliations:** 1Faculty of Electronics, Telecommunication and Information Technology, University “Politehnica” of Bucharest, 060042 București, Romania; 2Horia Hulubei National Institute for R&D in Physics and Nuclear Engineering (IFIN-HH), 077125 Măgurele, Romania

**Keywords:** digital receiver, obsolete analog radio, environment protection, recycling method, digital signal processor, radio frequency, vacuum tube, high fidelity

## Abstract

Since its inception, the electronics industry has mass-produced equipment. The fast evolution of electronic technologies made obsolete the entire generation of products and even technologies. Until the government issued regulations and guidelines on how to address the issue of reuse of obsolete electronic equipment, with special regard to the ones still operating (e.g., give it to family/friends, donate to charity, or sell to individuals or recycling companies), most of it was thrown out with usual rubbish, with a destructive effect on the environment. This paper presents the design techniques and methods for revaluation of obsolete vacuum tube analog receivers, with a focus on the manufacturing steps of a high-performance receiver. The choice of receiver type is not accidental at all, since tube technology is still a real success among audiophiles many providers offer vacuum tube amplifiers at considerably high prices. The redesign implied the original FM unit replacement with a DSP-based AM/FM tuner while the AM RF vacuum tube section has been preserved with the original architecture to allow the reception of the broadcast stations for the long-wave band and the alternative operation with the silicon tuner for the medium-wave and short-wave bands. The electrical performances of the modified receiver in terms of reliability, sensitivity, selectivity, and distortions on the reception chain are clearly superior to the original one, while the power consumption of the RF section is reduced more than 10 times from 11.5 W–15.5 W to 1 W. Last, but not least important, the proposed solution implied the use of few additional parts and resources and extended significantly the lifetime of the original vacuum tubes receiver. The work has been developed to serve as an example of how obsolete electronic equipment can be redesigned and reused avoiding its complete recycling or even worse, its disposal with usual rubbish. It has been imagined and performed as the initial step in launching a professional student contest on the reuse/redesign of obsolete equipment aimed at raising awareness regarding the issue of pollution with e-waste amongst students from the electronic departments of Romanian technical universities.

## 1. Introduction

The last 100 years in the history of mankind are sprinkled with events that have marked the lives of billions of people. Of these, accelerated technological development has allowed and encouraged over time people to know a better, lighter, and longer life than in the past. All this was achieved at a high price: the pollution of nature and the consumption of Earth’s resources [1]. It has now arrived that resources consumed by humanity in a year should be almost twice as high as they can produce for the earth at the same time [2]. It is clear that the direction in which mankind has grasped is not sustainable. In addition, if we look back in the past to make a comparison between the simple way of life of the inhabitants of a city in the XIXth century and the life of citizens who live in the same city today, we will find that the resources consumed, and environmental pollution have seen an exponential increase [3]. Unfortunately, nowadays, comfort means consumption—most of the time unnecessary—of natural resources. For this feeding, in a short period of time, a lot of industries, and large consumers of natural resources, were created. Among these, the consumer electronics industry occupies a leading place. She is also responsible for the massive pollution of soil, water, and air, things that affect the health of the population on all continents [4]. Pollution is generated by both the production process of electronic equipment and electronic waste (e-waste), both with an upward trend over time. Even some countries such as the UK addresses the issue of waste disposal of electronic equipment within the last decade [5] and public bodies, as well as private companies, offer guidelines on how to dispose of e-waste including recycling services, still in many parts of the world e-waste is a bin with usual rubbish. According to the Global E-waste Statistics Partnership—GESP (globalewaste.org), the Norwegians have the largest annual average amount of e-waste generated per household—57 kg, while the UK is in second place with 55 Kg. GESP aimed at monitoring the developments of e-waste over time and supporting countries to produce e-waste statistics and in 2022 launched a consultation on the Methodology for Measuring the Global Progress for e-Waste Legislation [6].

To better understand the current situation, we should look back in time to see what the goals of the electronics industry were in the second half of the XXth century. During this period, the main purpose of the industry was to deliver the most efficient and reliable equipment possible to the end customer [7]. One main and very useful characteristic for specialists was that complete technical documentation was made available to the public. The factories produced a reduced number of parts, many of them with the same electrical characteristics, so that using a small range of “universal” parts, almost any type of repair could be performed, no matter if the manufacturer was Japanese, American, or European. By using the through-hole technology, the repair process was shortened and made easier. There were also a large number of service centers, and the qualification of technicians was high. As factories faced a shortage of orders, the price of semiconductors was declining, and competition increased, the industry changed the policy, moving from the manufacture of high-quality products, with a long service life and which can be repaired, to cheaper products, with a shorter service life (ideally as long as the warranty lasts), to allow the sale of a large number of products at short intervals and to ensure orders in large quantities for factories. Thus, many products of poorer quality have appeared on the market, without solid technical support from the manufacturer. Moreover, as the technical documentation was completely missing in most cases, the repair of the products under warranty or out of warranty was almost impossible to perform. Additionally, if the past electronic equipment had a modular architecture to allow easier service by replacing defective modules, most equipment manufactured in the XXIst century has eliminated this principle, by concentrating all electronic components on a single PCB. As a consequence, the policy stating that any product that fails during the warranty period is replaced with a new one or equivalent became common practice, the service of the product switched from repairing individual failure device(s)/circuit(s) towards replacing the entire board or even the whole equipment transforming the defective product in e-waste.

On the other hand, the fast evolution of electronic technologies made obsolete entire generation of products and even technologies. Until the government issued regulations and guidelines on how to address the issue of reuse of obsolete electronic equipment, with special regards to the ones still operating (e.g., give it to family/friends, donate to charity, or selling to individuals or recycling companies), most of it was thrown out with usual rubbish as mentioned above.

All the above-mentioned facts are very common today, and the decision makers should promote policies that encourage the recycling as much as possible in parallel with the production of equipment with a higher lifetime, that can be repaired, and also reused/redesign even when the technology become obsolete.

This work presents a method by which an obsolete radio broadcast receiver based on vacuum tube technology can be transformed into a modern device with superior electrical performances, by using a small number of additional components. The choice of receiver type is not accidental at all, since the tube technology is still a real success among audiophiles many providers offer vacuum tube amplifiers (for example Luxman [8] or McIntosh [9], with considerably high prices [10]. Thus, the environment is protected because it avoids throwing the device in the bin, while the end customer can enjoy a high-quality audio receiver with a longer service life that can also be maintained over time with low costs for many years.

## 2. The Original Architecture of the Rossini 6002 Superheterodyne Receiver

The equipment chosen for this work is the Rossini 6002 superheterodyne receiver produced by VEB(K) Goldpfeil Elektroakustik Hartmannsdorf company from the German Democratic Republic (East Germany) in 1962 [11], which was sold in a large number of pieces in the block of Warsaw Pact member countries. The unit is designed to receive amplitude modulation (AM) broadcast stations in the long wave, medium wave, and three short wave bands, as well as frequency modulation (FM) broadcast stations in the OIRT band. Choosing this type of receiver was performed as in Romania there are RF power transmitters still in operation for all AM broadcast bands, and the evaluation of the operation of the receiver can be performed in any frequency range. The entire chassis with four speakers are mounted in a lacquered wooden box. All amplifying stages are built with vacuum tubes. The electrical characteristics of the RF section of the receiver are mentioned in Table 1 while the ones for the audio amplifier section are presented in Table 2 [12]. The shortwave band is divided into three sections to allow easy tuning of the frequency of the received radio station. The receiver has a high sensitivity in the medium wave range and is less sensitive to high frequencies in the higher range of the shortwave band.

Despite mounting the speakers in the same box, the audio section offers high-quality listening. Although the manufacturer’s documentation specifies that as the unit is a stereo receiver, the apparatus of our possession has not been equipped by the factory with a stereo decoder. Thus, the reception of the radio stations in the frequency modulation reception band (FM) in stereo mode is practically impossible, and a stereo audition can be carried out only by applying an audio stereo signal from an external device by selecting TA (turntable) or TB (tape recorder) inputs.

The audio section is also equipped with a five-position tone register, which can accentuate a certain frequency range, depending on the type of the desired listening program.

The block diagram of the original receiver provided by the producer as printed sheet inside the equipment (Figure 1) contains the following building blocks numbered from 1 to 11: the power supply unit (block 1), the audio amplifier (block 2), the 5-position tone register (block 3), a volume and bass control circuit (block 4), the IF module (block 5), one tuning indicator (block 6), the keyboard (block 7), a RF mixer and local oscillator for AM section (block 8), a RF preamplifier for the AM section (block 9), a ferrite rod antenna for long-wave and medium-wave bands (block 10), and a FM tuner (block 11).

The power supply unit contains an AC line filter (C_f1_, C_f2_), a power transformer (PT1), a selenium bridge rectifier (SR), and an RC filter (FILT). It is designed to provide AC filament voltage (6.3 V) and the DC bias voltages (anode, screen) for the vacuum tubes and the supply voltage for the backlight display scale light bulbs.

The input audio signal (from external source of from internal AM/FM detectors) is applied to a low frequency RC filter (block 4) formed by a bass cut circuit (BCC) and a stereo volume potentiometer. Then, the signal is applied to a preamplifier (SPREAMP) followed by a tone cut circuit (TCC) which provides the audio signal to the tone register unit (block 3) for further processing, and a fixed-gain stereo power amplifier (SPAA).

The reception of the amplitude modulation (AM) broadcast stations is accomplished by a tunable AM radiofrequency preamplifier (block 9) which increases the sensitivity of the receiver, a RF down conversion mixer (AM MIX) and a tunable local AM oscillator (block 8), and an intermediate frequency (IF) amplifier and detector (block 5). The frequency modulation radio signals are processed by a FM tuner (block 11) built from a tunable RF amplifier and an FM self-oscillating mixer, followed by the FM IF amplifier and detector (block 5). The strength of the received RF signal is displayed by using a magic-eye tuning indicator (block 6).

## 3. The Proposed Architecture

The block diagram of the modified receiver containing 13 blocks is presented in Figure 2, where each block is numbered and the newly added or changed elements have been thickened. Compared to the original scheme shown in Figure 1, the internal architecture of blocks 1, respectively, 10 to 13 was modified. To increase the service life of the receiver, the selenium rectifier is replaced by a silicon bridge rectifier inside the power supply unit (block 1). A high voltage regulator for anode bias is also added to increase the regulation coefficient and to prevent electrical noise intrusion into the sensitive amplifying stages from the AC power line. A supplementary low voltage power supply unit (LVPS, 13) provides the supply voltages for newly introduced circuits. The audio section (blocks 2 ÷ 4) retains its initial architecture. An analog input switch (block 12, InSW) is introduced on the signal path to allow the selection of multiple signal audio sources (external or internal) to be applied to the audio amplifier input. All initial radio frequency circuits used for the reception of the FM broadcast stations (Figure 1) have been completely removed or deactivated. The original FM unit is replaced by a DSP-based AM/FM tuner (block 11, Figure 2). An antenna signal input switch (ASW) allows the RF signal to be redirected to the silicon tuner (11) or to the AM RF vacuum tube preamplifier. The AM RF vacuum tube section (blocks 5 to 8 and block 9) has been preserved with the original architecture to allow the reception of the broadcast stations for the long-wave band and the alternative operation with the silicon tuner for the medium-wave and short-wave bands. Some switching sensors (block 11) are mounted inside the keyboard to permit the control of the additional installed blocks.

## 4. Results

The redesign of the receiver started with the evaluation of the performances of the audio section, which consists of three amplifying stages for each channel (Figure 3). For each audio channel, three amplifying stages based on vacuum tubes are used. The first stage, which also contains the volume and tone controls, is a tube amplifier circuit without negative feedback. For this reason, the frequency response of the audio amplifier is strongly non-linear. The audio output amplifier (blocks 2 to 3) presents negative feedback.

The first amplification stage is represented in Figure 4 only for the left channel. It uses a vacuum triode preamplifier in the common-cathode configuration. Two passive filters are mounted to the input and to the output of the preamplifier. The input filter consists of the bass cut (*R*_63_, *R*_65_, *C*_111_) and volume adjust (*R*_71_, *R*_67_-*C*_113_, *C*_115_, *R*_69_-*C*_117_, *R*_73_-*C*_119_) circuits. At the output, a treble cut circuit is connected (*C*_127_-*R*_81_).

The voltage gain [13] of the triode amplifier is:(1)$$A_{V}=\frac{\mu \cdot R_{79}}{R_{79}+r_{a}}=\frac{100\cdot 100k\Omega }{100k\Omega +62.5k\Omega }=61.53\approx 35dB$$

The preamplifier input circuit (*C*_121_-*R*_75_) is a high-pass filter with a −20 dB/decade slope and a lower −3 dB point that is determined by the equation:(2)$$f_{-3dB\_in\_p}=\frac{1}{2\cdot \pi \cdot R_{75}\cdot C_{121}}=6.36Hz$$

The tube input capacitance is [13]:(3)$$C_{in}=C_{gk}+C_{ga}\cdot (A_{V}+1)\approx 150pF$$

The volume adjustment is performed by a linear type of potentiometer with three taps (*R*_71_). An equivalent circuit for the filter formed with the bass cut and the volume potentiometers is depicted in Figure 5, where the multi-tap potentiometer was replaced by four potentiometers without taps. The audio signal is picked up from one (*V*_1_, *V*_2_, *V*_3_ or *V*_4_) of the four cursors of potentiometers (*R*_71-1_ ÷ *R*_71-4_) at a moment of time, depending on the cursor position (SET) of the volume potentiometer (*R*_71_, Figure 4).

In Figure 6, the RC filters are replaced by their equivalent impedances, and then the voltage transfer function of the circuit is determined depending on these impedances and the cursor position discrete values (SET∈{0;0.25;0.5;0.75}).

The voltage transfer function of the circuit presented in Figure 6 is a complex function that is difficult to calculate. To determine it for any frequency value, a simulation model for the multi-tap potentiometer is created by using the PSpice Model Editor [14]. A spice circuit file containing the first audio amplifying stage (Figure 4) is created (see Appendix A) and a simulation is performed. By modifying the values of the *R*_65_ (bass cut), *R*_71_ (volume), and *R*_81_ (treble cut) potentiometers, the effect of changing these values on the voltage gain of the preamplifier as a function of frequency is represented in Figure 7, Figure 8 and Figure 9. The potentiometer slider is varied in 10 equal steps between the maximum value and the minimum value.

The five-keys tone register consists in several RC filters. Its functions are depicted in Table 3.

Each key can be operated independently, so that a complex filtering function can be created by pressing one or more keys. The register is in the STEREO position (Figure 10). Other filters may be activated by pressing the corresponding key.

The spice simulation of the voltage transfer function of the filters mounted inside the tone register is represented in Figure 11. The slider of the treble cut potentiometer is set to halfway.

The power audio amplifier (left channel, Figure 12) has a series-shunt negative feedback topology. It consists of two amplification stages: a triode un-bypassed common cathode amplifier, followed by a pentode common cathode amplifier.

The negative feedback network (Figure 13) determines the voltage gain of the power amplifier. The balance function between the audio channels is performed by using a potentiometer (*R*_115_) mounted inside the negative feedback loop. The voltage transfer function of the negative feedback circuit is:(4)$$\underline{f_{V}}=\frac{\underline{V_{if}}}{\underline{V_{of}}}=\frac{R_{97}\cdot \left(R_{201}+R_{115}^{L}\right)}{R_{113}\cdot \left(R_{97}+R_{99}+R_{201}+R_{115}^{L}\right)+\left(R_{201}+R_{115}^{L}\right)\cdot \left(R_{97}+R_{99}\right)}\in \left[0.0147;0.1\right]$$

The voltage gain of the PA in the middle of audio frequency band is:(5)$$\underline{A_{V}}=\frac{\underline{a_{V}}}{1+\underline{f_{V}}\cdot \underline{a_{V}}}=\frac{1}{\frac{1}{\underline{a_{V}}}+\underline{f_{V}}}\approx \frac{1}{\underline{f_{V}}}\in \left[10;68\right]$$

It has a value of 13 for the default position (halfway) of the balance potentiometer.

For complete signal analysis throughout the audio transmission chain, the setup mentioned in Figure 14 is performed. A sinewave signal provided by the IEMI E0503 [15] generator is applied to the input of the audio amplifier. The frequency value is precisely monitored by the Philips PM6676 [16] counter. At the output, a digital multimeter (Keithley 197) [17] connected in series with the speakers displays the current flowing through the speaker’s coils. The oscilloscope [18] is used to verify that the shape of the signals at the input and output of the amplifier is sinusoidal. The universal level meter (UPM550) [19] measures the output voltage or the *THD* depending on user preferences.

First, the frequency response of the audio section is considered. A constant-amplitude variable-frequency signal is applied to the input of the amplifier. The universal level meter is set to measure the speaker’s AC RMS voltage. The results are depicted in Figure 15.

Second, the *THD* measurement of the amplifier is achieved. A filter board formed by three filters (Figure 16) which introduces a variable attenuation depending on the frequency (Figure 17) is built for this measurement according to the meter factory specifications [20].

The filter board is designed to introduce an attenuation of at least 66 dB [20] at the test frequency of 1000 Hz. After the filter board is mounted inside the UPM 550 Level Meter, the *THD* measurement is directly performed at various power levels. The *THD* value is displayed in dBv on the meter scale. To be converted to percentages (Table 4), the relation (block 5) is used:(6)$$THD\left(\%\right)=10^{\frac{THD(dBv)}{20}}\cdot 100$$

To reduce the noise induced by the electrical grid in the audio section and to increase the mean time between failures (MTBF), a supply circuit for anode biasing (Figure 18) is replacing the old selenium bridge rectifier. This circuit is made up of a simple pre-regulator formed by a voltage reference (Z_1_) and a power regulator with a field effect transistor (M1), followed by a negative-feedback voltage regulator in the structure of which is a voltage reference (Z2), an error amplifier (EA), a series pass element (SPT), and the short circuit protection circuit (R_s_, SCP). The regulator provides 262 V regulated voltage with a maximum output current of 150 mA, with line regulation less than 1%, load regulation less than 0.5%, and PSRR = 43 dB.

The anode power supply unit is mounted in the vicinity of the power transformer in the top area of the receiver chassis.

The last step in changing the architecture of the device is the redesign of the RF section. The original RF FM section of the receiver is completely removed. A mechanical-tuned analog display broadcast tuner (block 11, Figure 2) using a digital CMOS RF integrated circuit [21] replaces the RF section for medium-wave, short-wave, and FM bands (Figure 19) [22]. Moreover, the RF AM vacuum tubes section is still used to allow reception of stations in the long-wave band, and, also, to be an alternative receiver for the other AM bands due to a small advantage of the continuous (analog) tuning.

The RF switch (ASW) selects the RF signal received by the antenna to be sent between the DSP tuner or the RF tube preamplifier inputs. A ferrite rod antenna is used to receive radio stations in the medium-wave (by using the DSP tuner or the tube AM section) and the long-wave (tube AM section) bands in noisy reception environments.

The FM RF signal is passed through a high-pass filter (HPF) to be applied to a LC circuit (*L*_FM_, *C*_V1_) tuned on the desired receiving frequency.

The AM RF signal is injected to a RF preamplifier circuit through a low-pass filter (LPF). The load of the AM preamplifier is represented by the LC tuning circuit formed by L_11_^`^-C_V2_. Then, the RF signal is processed by the internal circuitry of the RF IC receiver.

A low noise amplifier (LNA) is used to increase the RF voltage level before it is applied to an in-phase and quadrature mixer. The mixer provides the RF signal to the inputs of two analog-to-digital converters which transform it in binary format. The digital signal is subsequently processed by the DSP unit and finally converted into analog audio signal with two digital-to-analog converters [22].

The tuning is performed in steps by using two external multiturn potentiometers (P_1_ for AM tuning, P_2_ for FM tuning) while the selections for receiver type (tube or DSP), band, or audio signal are made by using the keyboard (block 7). The tuning (block 6, Figure 2) and stereo indicators are turned on through the control interface.

A low voltage linear power unit (LVPS) connected to the filament winding of the AC mains transformer (PT1, Figure 2) provides the bias voltages for the digital tuner, indicators, potentiometers (P_1_, P_2_), and relays mounted inside switching blocks (ASW, InSw, Ferrite Rod Antenna, Figure 19). To simplify connections between modules, the tuner and a section of the low voltage power supply unit are arranged on the same circuit board.

The measurements for the RF block of the receiver are achieved by using the setup presented in Figure 20.

The first signal generator (Marconi 2023A) [23] provides the amplitude and frequency modulated signals. A second generator (Agilent N5171B) [24] will only output the carrier wave for the selectivity and image rejection measurements. The power combiner that sum-up the RF signals from the RF signal generators to be applied to the tuner unit is built using a commercial RF dedicated device [25]. The coaxial connecting cables are kept as short as possible. The insertion loss on the entire RF transmission path (RF signal generator, RF power combiner, AM/FM tuner) is about 6 dB.

Since the receiver is not placed in an anechoic chamber, the frequency at which the measurements are completed is chosen near the middle of the working band such that so that there is no disturbing signal received from a broadcasting station that transmits with the same frequency as the signal generators (blocks 1, 2). To determine it, a spectrum analyzer [26] equipped with a telescopic antenna measures the RF field around the receiver. The chosen frequency is the one at which the power of the RF signal indicated by the analyzer is minimal, to avoid interference with radio stations.

All RF measurements (Table 5) are made considering the environment in which these are completed, and the recommendations provided by the manufacturers of the RF DSP integrated circuit [27,28,29] and of the universal level-meter [19].

The oscilloscope monitors the shape and the frequency of the demodulated audio signal (1 kHz) for both channels (L, R). The level meter is configured to measure the amplitude of the audio signal, and the total harmonic distortion (*THD*) for each band.

Furthermore, the noise figure of the receiver is determined by using the “gain” method [30,31] adapted for the available spectrum analyzer. The setup is presented in Figure 21.

Two sets of measurements are performed. First, the RF signal generator provides modulated RF signal in frequency or amplitude to the input of the broadcast receiver. The spectrum analyzer is connected by turn to the external speaker terminals of the receiver to monitor the audio signal (*S*_o_) and to the signal generator to measure the power of the RF input signal (*S*_g_). Second, a non-inductive RF resistor (*R*_in_) is mounted to the input terminals of the receiver. The spectrum analyzer measures the output noise (*N*_O_) generated by the receiver on its speakers.

The noise figure is defined by:(7)$$NF=10\cdot \mathrm{lg}\frac{n_{o}}{g\cdot k\cdot T\cdot RBW}$$
where *n*_o_ is the output noise, *g* is the gain of the receiver, *k* is the Boltzmann’s constant, *T*—absolute temperature, and *RBW*—is the resolution bandwidth of the spectrum analyzer.

By breaking the fraction and performing numerical calculation at *T* = 290 K, the following relation results:(8)$$NF=10\cdot \mathrm{lg}(n_{o})-10\cdot \mathrm{lg}(g)+174dBm-10\cdot \mathrm{lg}(RBW)$$

By considering:(9)$$N_{0}=10\cdot \mathrm{lg}(n_{o})$$
(10)$$G=10\cdot \mathrm{lg}(g)=S_{o}-S_{g}$$

The Equation (7) becomes:(11)$$NF=N_{o}-G+174dBm-10\cdot \mathrm{lg}(RBW)$$

By considering *RBW* = 10 Hz with minimal value, for the available spectrum analyzer *NF* has the following expression:(12)$$NF=N_{o}-G+164dBm$$

The results are presented in Table 6.

## 5. Discussion

After all tests are performed, the manufactured subassemblies are mounted in the receiver. The keyboard (block 7), the analog input switch (InSw, block 12), and half of the low voltage power supply (LVPS, block 13) are mounted on the bottom of the chassis.

Under each key is glued a magnet that switches the reed relay when pressed. The tuning potentiometer for the AM section (AM_DSP_TUN) is placed near the variable air capacitor of the receiver. Furthermore, all aged paper and electrolytic capacitors are replaced with new others. The tuner unit (block 11) is mounted in the place of the original vacuum tube FM tuner. For the reception in the MW band using a ferrite antenna (block 10), an additional winding (L_11_) is added to the original coil (L_11_) to be adapted [28] to the input tuning circuit of the RF DSP integrated circuit.

By canceling the supply voltages for blocks 5, 8, and 9 (Figure 2), the power consumption in the RF section is considerably lowered. The power dissipation for the original RF section (Table 7) varies between 11.5 W and 15.5 W [32,33,34,35]. The designed low voltage power supply module (LVPS) delivers a maximum power of 1 W to the entire silicon RF unit, reducing the power consumption by more than 10 times.

By analyzing and comparing the data after the measurements are performed, the overall performance of the RF unit in terms of sensitivity and selectivity is substantially improved. Additionally, the receiver lifespan is significantly extended.

## 6. Conclusions

This work presents the techniques of modernizing and improving the architecture of an obsolete vacuum tube receiver and transforming it into high-quality audio equipment. In order to properly measure its parameters, a test bench has been specially designed and built. The electrical performances of the modified receiver in terms of reliability, sensitivity, selectivity, and distortions on the reception chain are clearly superior to the original one, while the power consumption of the RF section is reduced more than 10 times from 11.5 W–15.5 W to 1 W. Last, but not least important, the proposed solution implied the use of few additional parts and resources and extended significantly the lifetime of the original vacuum tubes receiver.

The work has been developed to serve as an example of how obsolete electronic equipment can be redesigned and reused avoiding its complete recycling or even worse, its disposal with the usual rubbish. Moreover, the choice of receiver type is not accidental at all, since the tube technology is still a real success among audiophiles and many providers offer vacuum tube amplifiers at considerably high prices.

Such approaches should be further promoted considering the raw materials shortage and the need of lowering pollution by lowering the level of e-waste produced worldwide. In this respect, the decision-makers should start to more actively propose and implement policies that encourage recycling as much as possible in parallel with the production of equipment with a higher lifetime, that can be repaired, and also reused/redesigned even when the technology becomes obsolete.

This work has been imagined and completed as the initial step in launching a professional student contest on the reuse/redesign of obsolete equipment aimed at raising awareness regarding the issue of pollution with e-waste amongst students from the electronic departments of Romanian technical universities.

## Figures and Tables

**Figure 1 sensors-22-06784-f001:**
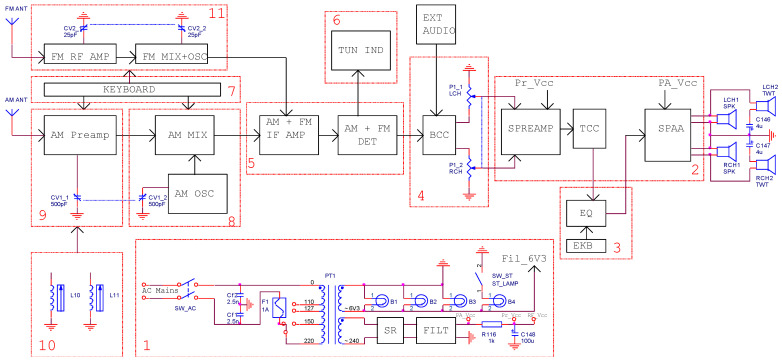
The block diagram of the original vacuum tube receiver.

**Figure 2 sensors-22-06784-f002:**
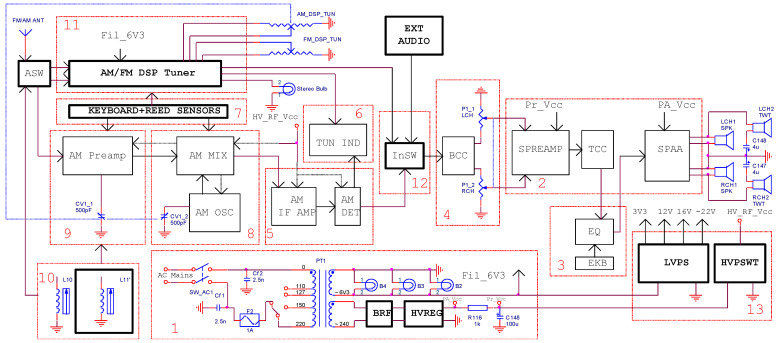
The proposed architecture of the broadcast receiver.

**Figure 3 sensors-22-06784-f003:**
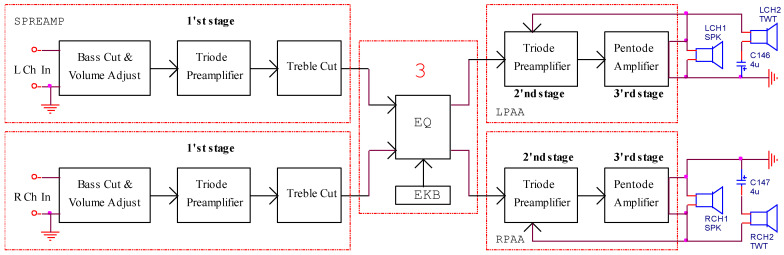
The structure of the modified audio amplifier.

**Figure 4 sensors-22-06784-f004:**
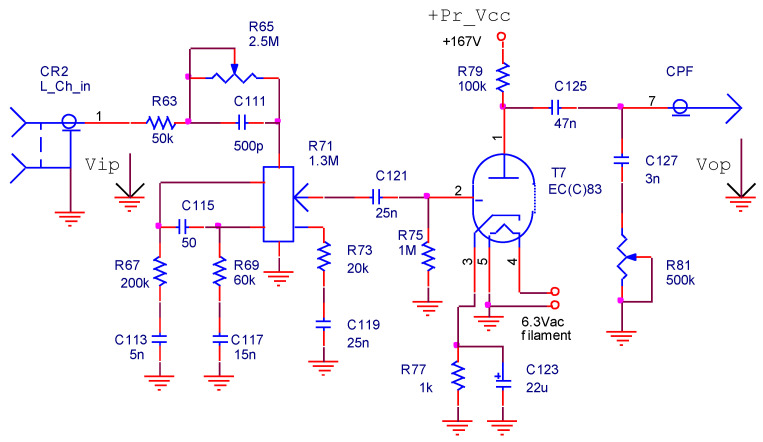
The first stage of the audio amplifier.

**Figure 5 sensors-22-06784-f005:**
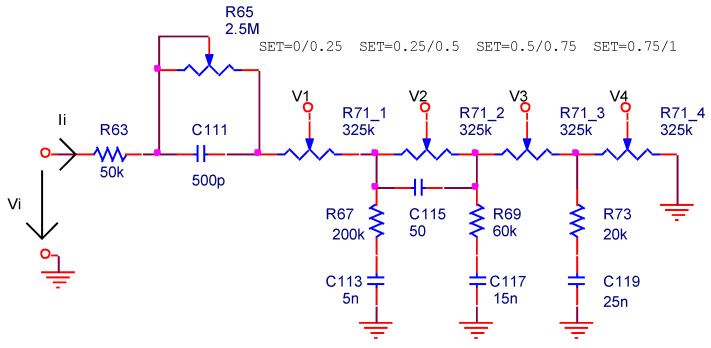
The replacement of the multi-tap potentiometer by an equivalent circuit.

**Figure 6 sensors-22-06784-f006:**
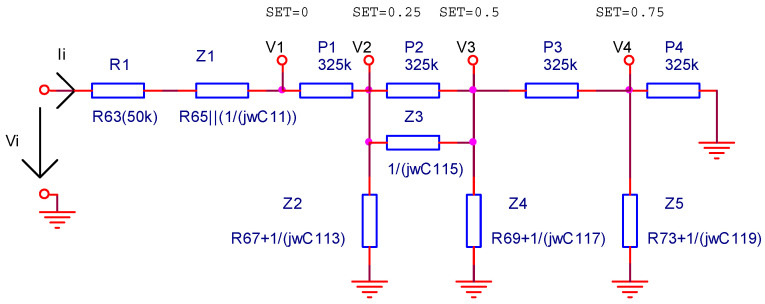
The structure of the input filter by considering the impedances of the RC filters and four discrete values for the cursor of the volume potentiometer.

**Figure 7 sensors-22-06784-f007:**
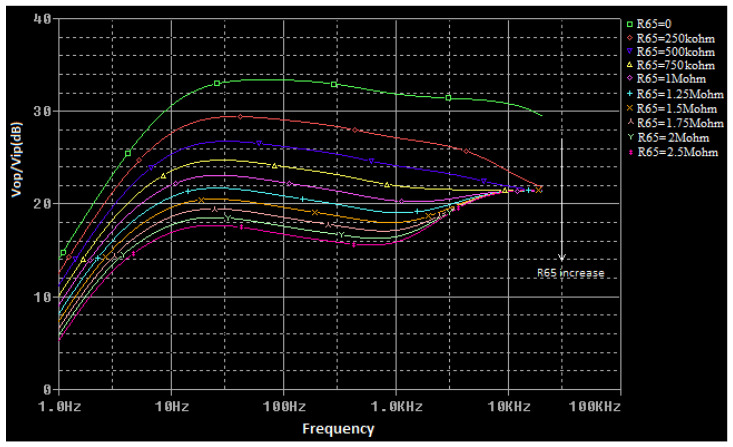
The influence of the *R*_65_ (bass cut) potentiometer in the voltage transfer function of the audio preamplifier (*R*_71_, *R*_81_ have the maximum values).

**Figure 8 sensors-22-06784-f008:**
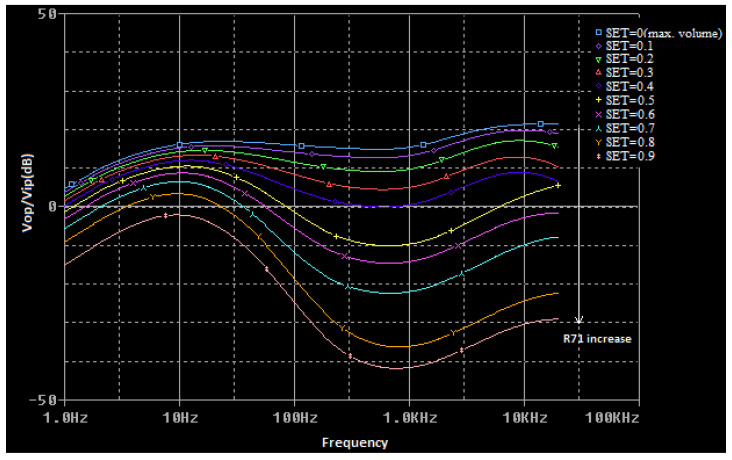
The influence of the *R*_71_ (volume) potentiometer in the voltage transfer function of the audio preamplifier (*R*_65_, *R*_81_ have the maximum values).

**Figure 9 sensors-22-06784-f009:**
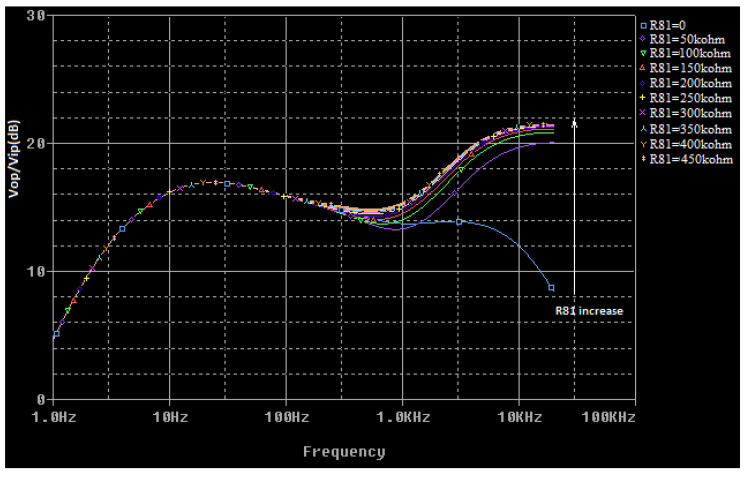
The influence of the *R*_81_ (treble cut) potentiometer in the voltage transfer function of the audio preamplifier (*R*_71_, *R*_65_ have the maximum values).

**Figure 10 sensors-22-06784-f010:**
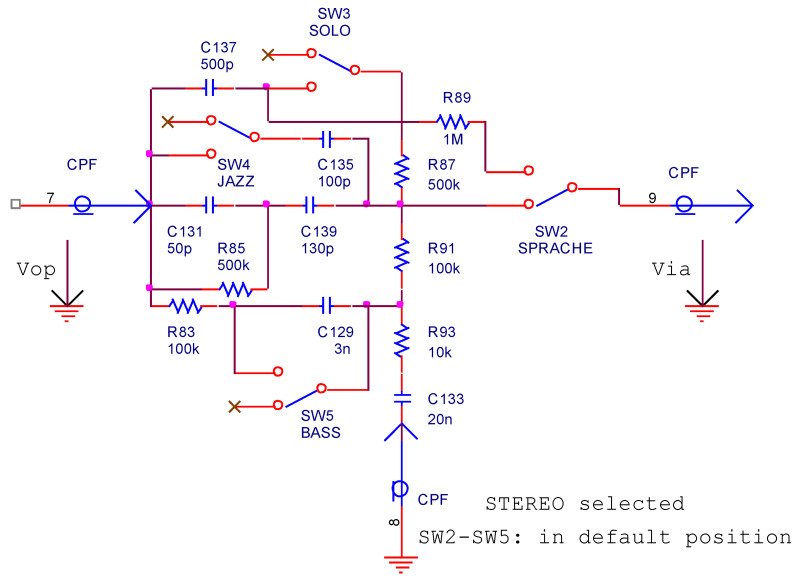
The tone register circuit.

**Figure 11 sensors-22-06784-f011:**
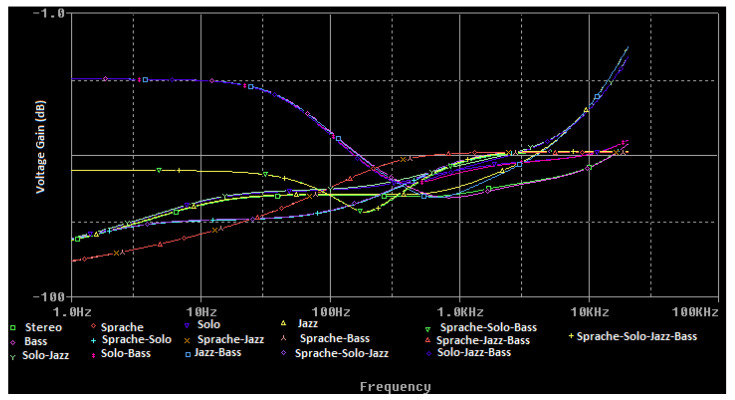
The frequency response of the tone register filters.

**Figure 12 sensors-22-06784-f012:**
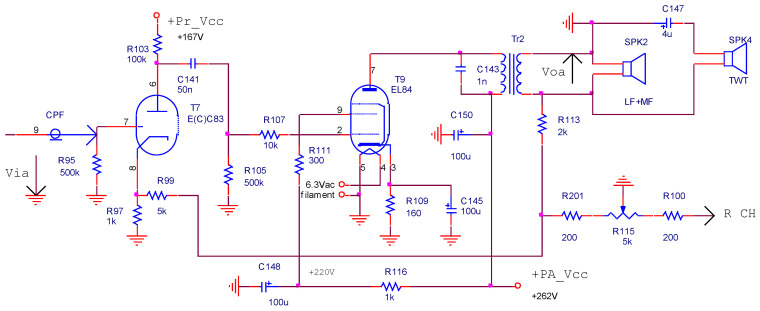
The power audio amplifier (2nd and 3rd amplifying stages).

**Figure 13 sensors-22-06784-f013:**
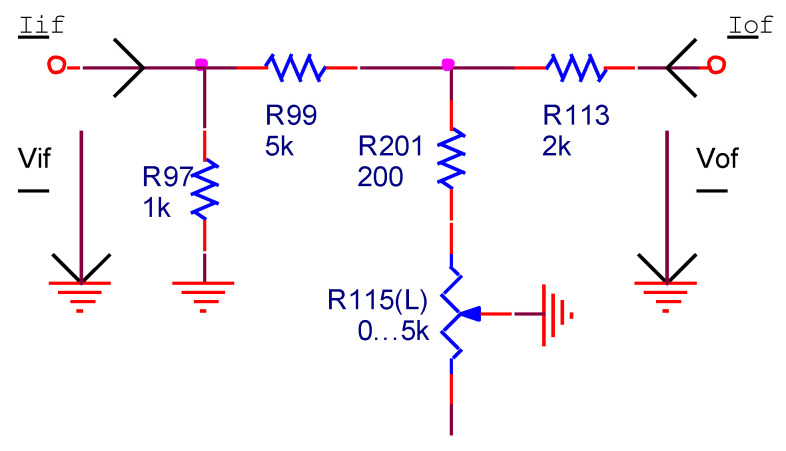
The PA negative feedback network.

**Figure 14 sensors-22-06784-f014:**
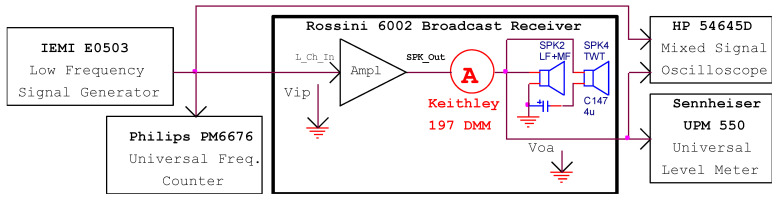
The measurements setup for the audio amplifier section of the broadcast receiver.

**Figure 15 sensors-22-06784-f015:**
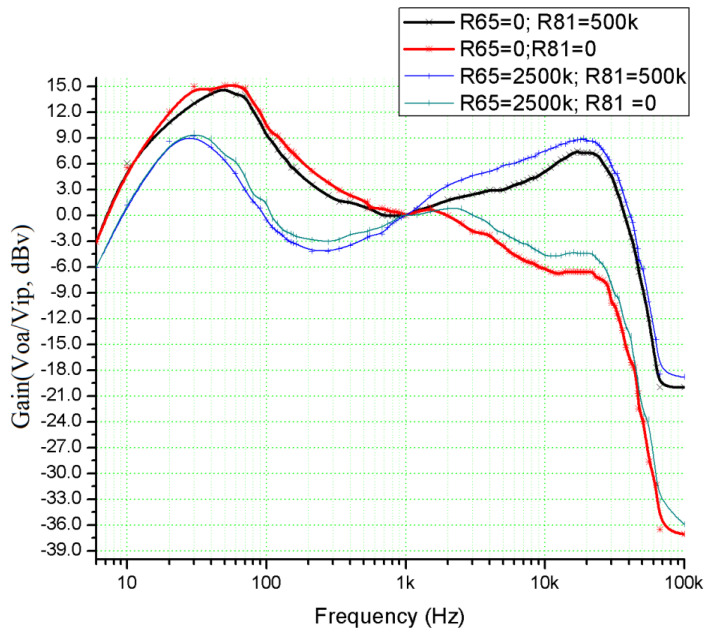
The amplitude–frequency characteristic of the audio amplifier for various values of *R*_65_ (bass–cut) and *R*_81_ (tone–cut) potentiometers (Stereo mode, sine wave input signal, 10 Hz ÷ 100 kHz range, 0 dBv voltage).

**Figure 16 sensors-22-06784-f016:**
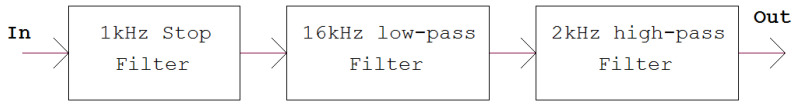
The block diagram of the filter board.

**Figure 17 sensors-22-06784-f017:**
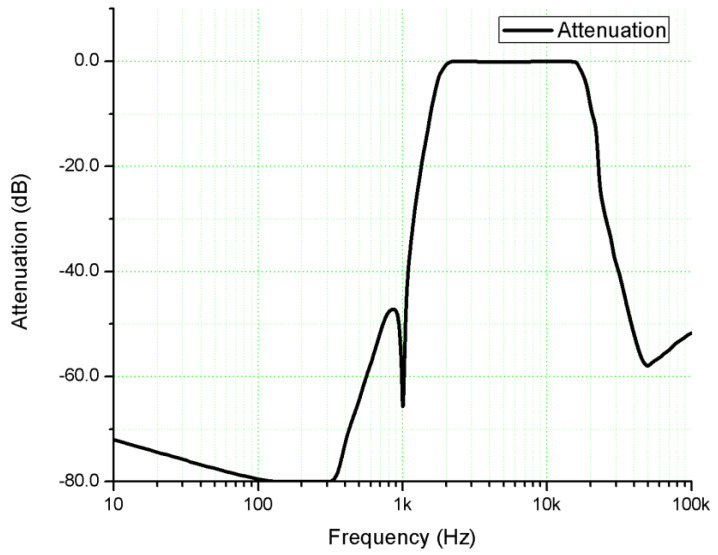
The frequency response introduced by the filter board used for *THD* measurements (input signal: sine wave, range 10 Hz–100 kHz, amplitude 0 dBv).

**Figure 18 sensors-22-06784-f018:**
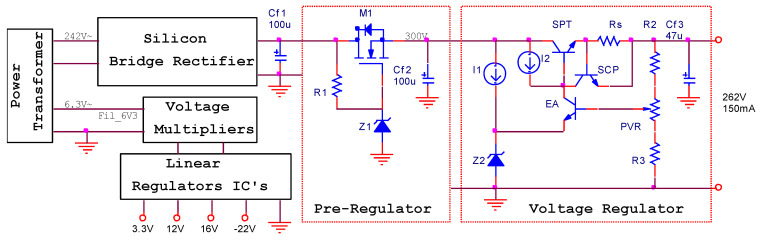
The block diagram for the power supply section.

**Figure 19 sensors-22-06784-f019:**
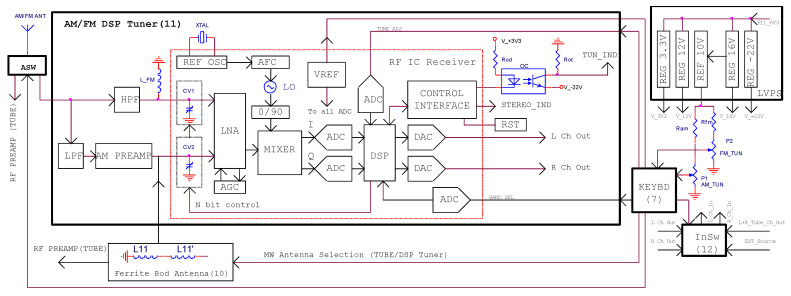
The structure of the tuner section.

**Figure 20 sensors-22-06784-f020:**
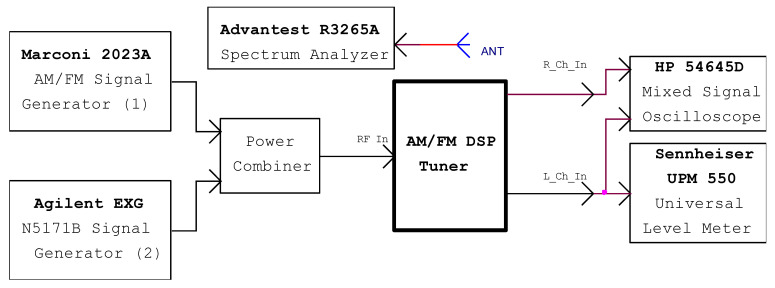
The measurement setup for the digital RF receiver.

**Figure 21 sensors-22-06784-f021:**
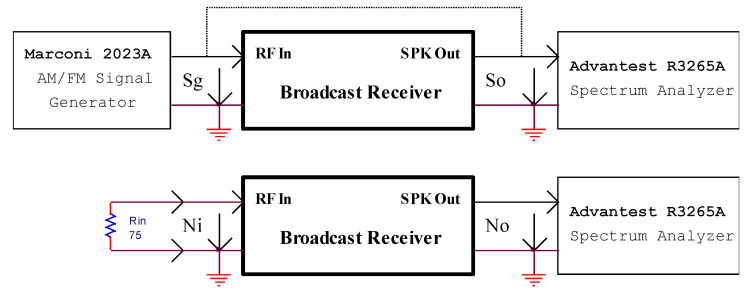
The setup for noise figure measurement.

**Table 1 sensors-22-06784-t001:** Technical RF characteristics of the Rossini 6002 broadcast receiver.

Band	Frequency Range ^1^	Sensitivity ^2^ (µV)	Selectivity (dB)	Image Rejection Ratio (dB)
Long wave (LW)	150 kHz–410 kHz	10–15	-	39
Medium wave (MW)	510 kHz–1630 kHz	8	53 ^3^; 75 ^4^	40–54
Short wave 1 (K1)	2 MHz–6 MHz	8–10	-	28–31
Short wave 2 (K2)	5.9 MHz–12.2 MHz	20–25	-	21–24
Short wave 3 (K3)	14 MHz–22 MHz	25–35	-	12–17
FM band (UKW)	66 MHz–72 MHz	16	49 ^5^	-

^1^ The intermediate frequency is 478 kHz for the AM bands and 10.7 MHz for the FM band. ^2^ Sensitivity is measured for an output power of 50 mW and for a signal-to-noise ratio of 26 dB. ^3^ The measurement is performed at 1 MHz when the receiver is tuned to a local radio station (short distance between the receiver and the AM station). ^4^ The measurement is performed at 1 MHz when the receiver is tuned to a far radio station (long distance between the receiver and the AM station). ^5^ The measurement is performed at 69 MHz for a ±300 kHz detuning.

**Table 2 sensors-22-06784-t002:** The characteristics of the audio amplifier section.

Frequency Range (Hz) ^1^	Crosstalk (dB)	Output Power (W)	Speakers
300–160,000	35	2 × 5 ^2^	2 × 6 VA, 7 Ω ^4^
		2 × 4 ^3^	2 × 1.5 VA, 12 Ω ^5^

^1^−3 dB band. ^2^ This value is measured for *THD* = 10%. ^3^ This value is measured for *THD* = 7%. ^4^ Full-range elliptical loudspeakers mounted on the front side of the receiver box. ^5^ Circular shape tweeters mounted on the left side and on the right side of the receiver box.

**Table 3 sensors-22-06784-t003:** The tone register positions.

Function	Key	Components Introduced
STEREO	-	-
SPRACHE	SW2	*R*_89_-*C*_137_
SOLO	SW3	*R*_87_-*C*_137_
JAZZ	SW4	*C* _135_
BASS	SW5	*C*_129_ s.c.
SPRACHE-SOLO	SW2-SW3	*R*_87_, *R*_89_, *C*_137_
SPRACHE-JAZZ	SW2-SW4	*R*_89_, *C*_135_, *C*_137_
SPRACHE-BASS	SW2-SW5	*R*_89_, *C*_137_, *C*_129_ s.c.
SOLO-JAZZ	SW3-SW4	*R*_87_, *C*_135_, *C*_137_
SOLO-BASS	SW3-SW5	*R*_87_, *C*_137_, *C*_129_ s.c.
JAZZ-BASS	SW4-SW5	*C*_135_, *C*_129_ s.c.
SPRACHE-SOLO-JAZZ	SW2-SW3-SW4	*R*_87_, *R*_89_, *C*_135_, *C*_137_
SPRACHE-SOLO-BASS	SW2-SW3-SW5	*R*_87_, *R*_89_, *C*_137_, *C*_129_ s.c.
SPRACHE –JAZZ-BASS	SW2-SW4-SW5	*R*_89_, *C*_135_, *C*_137_, *C*_129_ s.c.
SOLO–JAZZ-BASS	SW3-SW4-SW5	*R*_87_, *C*_135_, *C*_137_, *C*_129_ s.c.
SPRACHE-SOLO-JAZZ-BASS	SW2-SW3-SW4-SW5	*R*_87_, *R*_89_, *C*_135_, *C*_137_, *C*_129_ s.c.

**Table 4 sensors-22-06784-t004:** *THD* measurement ^1,2,3,4^.

Output Power (W)	*THD* (%)
1 m	0.01
10 m	0.04
20 m	0.05
50 m	0.1
100 m	0.16
150 m	0.19
200 m	0.25
250 m	0.31
500 m	0.67
1	1.12
1.5	2.24
2	2.81
3	4.7
4	6.62
5	8.91
7	17

^1^ Sine wave signal from generator applied to TB input (*V*_in_= 0 dBv, *f*_in_ = 1000.03 Hz). ^2^ Original vacuum tubes (manufactured 1962) mounted inside the amplifier. ^3^ Worst case scenario (maximum distortion values obtained with *R*_65_ = 0 Ω; *R*_81_ = 500 kΩ); ^4^ The amplifier power level is obtained by adjusting the cursor position of *R*_71_.

**Table 5 sensors-22-06784-t005:** The RF parameters of the modified receiver.

Band	Frequency Range	Sensitivity ^4^ (µV)	Selectivity ^10^ (dB)	Image Rejection (dB)	*THD* ^11^
Medium wave (MW) ^1^	522 kHz–1620 kHz	8 ^5^	56 ^5^	118 ^5^	0.2
Short wave 1 (K1) ^2^	3.2 MHz–7.6 MHz	2 ^6^	40 ^6^	52 ^6^	0.3
Short wave 2 (K2) ^2^	5.9 MHz–18 MHz	4 ^7^	39 ^7^	87 ^7^	0.3
Short wave 3 (K3) ^2^	9 MHz–22 MHz	10 ^8^	38 ^8^	97 ^8^	0.3
FM band (UKW) ^3^	87 MHz–108 MHz	2.5 ^9^	64 ^9^	47 ^9^	0.1

^1^ Channel spacing is 9 kHz. ^2^ Tuning step is 5 kHz. ^3^ Deemphasis is 75 µs and tuning step is 100 kHz. ^4^ Sensitivity is measured for an output power of 50 mW and for a signal-to-noise ratio of 26 dB. ^5^ The measurement is performed at f = 1 MHz. ^6^ The measurement is performed at f = 5.15 MHz. ^7^ The measurement is performed at f = 11.2 MHz. ^8^ The measurement is performed at f = 16.1 MHz. ^9^ The measurement is performed at f = 97.4 MHz. ^10^ 9 kHz detuning for MW and SW bands, and ±200 kHz detuning for FM band. ^11^ The level of the audio signal is in the 50 mV ÷ 90 mV range.

**Table 6 sensors-22-06784-t006:** The noise figure measurements for the modified receiver.

Band	Frequency	*S*_g_ (dBm) ^3^	*S*_o_ (dBm) ^3,4^	*N*_o_ (dBm/Hz) ^3,5,6^	*NF* (dBm)
Medium wave (MW) ^1^	1035 kHz	−92.3	−6.9	−61.09	17.5
FM band (UKW) ^2^	97.4 MHz	−96.2	−5.7	−58	15.5

^1^ Signal generator settings: AM, *f*_mod_ = 1 kHz, AM depth = 30%; ^2^ signal generator settings: FM, *f*_mod_ = 1 kHz, Δ*f* = 22.5 kHz; ^3^ spectrum analyzer settings: CF = 1 kHz, VBW = 1 Hz, SWP = 10 s; ^4^ receiver audio output power is 60 mW per channel; ^5^ all audio amplifier knobs remain in the same position during the measurement; ^6^ squelch function deactivated (N/A).

**Table 7 sensors-22-06784-t007:** The power consumption of the RF vacuum sections.

Tube	Power Consumption (W)
ECC85 ^FM Tuner^	4
EF89 ^FM IF amp/AM RF preamplifier^	2.5
ECH81 ^FM IF amp/AM mixer and oscillator^	3
2 × EBF89 ^IF amplifier^	2 × 3

Total 15.5 W (FM mode), 11.5 W (AM mode).

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
