# Peer review of "High Performance Broadcast Receiver Based on Obsolete Technology"

_sensors, 2022, doi:10.3390/s22186784_

Round 1

Reviewer 1 Report

1. Though the title is good and understandable, the word obsolete is used in the title, even though the paper is about old equipment, so I suggest keeping the word obsolete in the title.

2. The abstract does not present any results and conclusions. It would be better to include a summary of the results performance together with a conclusion and future work in the abstract

3. In the introduction, I do not see any references to a literature review on Obsolete. I would like you to write a review on it that is precise and accurate. 

4. As shown in [7-12] and [12-13], the references in the introduction would be better spread out to the specific sentence associated with each reference, and don't repeat the reference once it has been used.

5. The electrical performance of the audio amplifier, which is described in Table 2, is not clear from the information provided. It should be based on the characteristics of the object

6. The frequency ranges in table 1 are given, but there is no reference, so please provide a reference for the ranges and make them specific

7. It can be seen that Figure-1 is the intended block diagram of the vacuum tube receiver, but how it gets linked with the obsolete receiver is not shown.  

8. A similar connection must be made between Figure-2 and the obsolete figure  

9. Despite the fact that the results are very clear, the question remains as to what frequency is targeted and why it differs from other frequencies

10 The figure 16, the figure 20, the figure 22 and the figure 23, are not clear with the title of the paper, too many implementation figures, must be specific to the paper.

11. In the conclusion, the authors state "The power consumption of the RF section has been considerably reduced." but they do not specify how much has been reduced.

12. I would also like to suggest that in the conclusion, you include the conclusion as well as future work, just as the abstract suggests

13. It is also necessary to update the reference as well

Author Response

Dear Evaluator,

thanks for your remarks and suggestions that we carefully address within the updated version of the paper.

Reviewer 2 Report

Since its inception, the electronics industry has produced a lot of equipment. As technological advancement progressed rapidly, its products soon became technically obsolete, with the usual practice being to dispose of them directly in trash, with a destructive effect on the environment. This article presents design techniques and methods for revaluation of old equipment, with application on the manufacturing steps of a high-performance broadcast digital receiver built on an existing architecture of an old analog receiver.

Comments:

1-      What is your novelty?

2-      The abstract is poor. Please improve it.

3-      Why does the gain has around 25dB variation in Figure 15?  

4-      Please report the Noise Figure.

5-      Please compare this work with others.

Author Response

Dear Evaluator,

thanks for your remarks and suggestions that we carefully address within the updated version of the paper and detailed within the attached doc. Hope they met your requirements.

Your sincerely

Round 2

Reviewer 2 Report

Hello,

The new version can be published in Sensor Journal.